# AdaHC: Accelerating Multi-Token Prediction with Adaptive Head Chunking with Pipeline Parallelism

**Yan Wang** [1] [*]  **Chang Si** [2]  **Kaiming Yang** [3]  **Zhipeng Zhang** [2]  **Weijian Liu** [1]  **Man Yuan** [2]  **Mingzhen Li** [1]
**Yong Li** [2]  **Weile Jia** [1]

## Abstract

Multi-token prediction (MTP) architecture is widely adopted in LLMs. MTP blocks can be appended to the tail of model to predict additional tokens. However, when training with pipeline parallel, MTP leads to more pipeline bubbles and deteriorates the pipeline efficiency. Based on in-depth analysis of MTP architectures and loss functions, we have identified the parallel nature of the MTP blocks, and leverage it for superior pipeline scheduling. We propose AdaHC, an adaptive pipeline scheduling framework for accelerating LLMs training with MTP block(s). AdaHC splits the output heads into chunks and reassembles the chunks to generate balanced pipeline stages, and performs adaptive activation forwarding to preserve the numerical equivalence. Experimental results show that AdaHC improves the training throughput of SOTA LLMs with diverse MTP configurations by $1.35\times$ on average. This work paves a new direction for practical pipeline training.

## 1. Introduction

Multi-token prediction (MTP) architecture predicts multiple future tokens in parallel at once to densify the training signals and improve the data efficiency (Gloeckle et al., 2024; Liu et al., 2025b; Mahajan et al., 2025), which has been widely validated in state-of-the-art large language models (LLMs) such as DeepSeek-V3 (Liu et al., 2024) and Qwen3-Next (Qwen Team, 2025). The common implementation of MTP is using $D$ sequential MTP blocks (each block mainly contains an transformer block and an output head) after the

backbone to predict $D$ additional tokens (containing $D + 1$ output heads). Meanwhile, as one essential parallelization scheme for distributed LLM training, pipeline parallelism (PP) (Narayanan et al., 2019; 2021) partitions the model layers across devices and executes them in a pipeline manner, alleviating per-device memory bottlenecks while offering strong scalability via device-to-device communication. However, in hybrid-parallel training scenarios, naively applying PP to MTP can introduce substantial *pipeline bubbles*, significantly deteriorating training efficiency. The fundamental issue arises from load imbalance across pipeline stages introduced by MTP, in which several output heads are concentrated in the last pipeline stage. This concentration makes the final stage computationally more demanding than the preceding ones, resulting in a heavier "tail".

The pipeline bubbles caused by model architecture contain two categories: *1) Schedule bubble.* Assuming that the pipeline stages among devices are balanced, the schedule bubble arises from the inter-device data dependencies, and the fine-grained model partitioning can help reduce the schedule bubbles in warmup and cooldown phases (Zheng et al., 2022). *2) Imbalance bubble.* Because the pipeline stages among devices are usually not balanced, devices should wait for the slowest one (usually with more layers) for data production, causing periodic boundary stalls for every micro-batch (Dean & Barroso, 2013). However, MTP further deteriorates the pipeline bubble, as illustrated in Figure 1(a). Specifically, *1)* the output head for MTP is coupled with the transformer block, which restricts the partitioning granularity and hurdles the reduction of schedule bubble; and *2)* due to the additional output head, the tail stage with MTP is significantly slower than the other stages, leading to more imbalance bubble, and the imbalance is further amplified as the number of MTP blocks increases. Concretely, we denote the computing time of each transformer block by $T_t$, the computing time of each output head by $T_o$, the number of micro-batches by $\mathrm{mbs}$, and the number of output heads by $n$. In Figure 1(a), the imbalance bubble is $(\mathrm{pp\_size} - 1) \times \mathrm{mbs} \times 3 \times n \times (T_o - T_t)$, which severely constrains the pipeline throughput.

There are two methods tackling the tail-heavy issue un-

---

[*]Work done during an internship at Alibaba Group. [1]University of Chinese Academy of Sciences, Beijing, China [2]Alibaba Group, Beijing, China [3]National University of Singapore, Singapore. Correspondence to: Weile Jia <jiaweile@ict.ac.cn>, Mingzhen Li <limingzhen@ict.ac.cn>.

*Proceedings of the 43$^{rd}$ International Conference on Machine Learning*, Seoul, South Korea. PMLR 306, 2026. Copyright 2026 by the author(s).

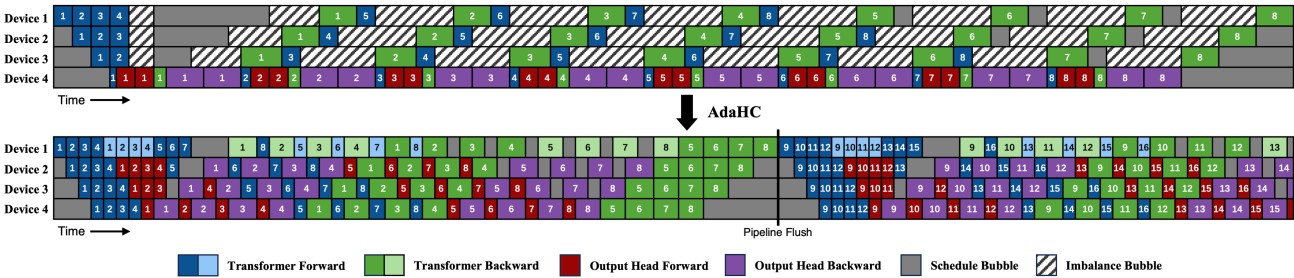

*Figure 1.* Pipeline schedules on 4 devices for an experimental model consisting of 9 transformer layers and one MTP block. Top: the default non-interleaved 1F1B schedule, incurs schedule bubbles and imbalance bubbles due to stage-time imbalance between transformer layers and the output-head computation. Bottom: AdaHC with interleaved 1F1B schedule, which balances stage workload to remove imbalance bubbles and minimizes unavoidable pipeline warmup/cooldown schedule bubble. For brevity, the execution time ratio of transformer layer and output head is set to 1:3, and the ratio of forward and backward is set to 1:2.

der a *single* output head, yet neither readily extends to the multiple output head scenario. The first is *layer redistribution* (Smith et al., 2022): moving additional transformer blocks to stages before the tail to hide the output head latency and mitigate inter-stage imbalance. However, with multiple output heads, it should compensate more transformer blocks to previous stages, which can rapidly increase the parameter memory and activation memory, breaking memory balance and potentially exceeding per-device capacity. The second is *vocabulary-wise parallelism* (Tsung et al., 2025; Li et al., 2025), which partitions the output heads and softmax operations across multiple devices, and then aggregates results via collective communication. If with MTP, the communication must be repeated for each output head, causing frequent pipeline stalls.

Based on in-depth analysis of MTP architectures, we find out *the parallel nature of the MTP heads* for reducing the pipeline bubbles. *1) No data dependence across output heads.* The overall MTP loss $\mathcal{L}_{\text{MTP}}$ is the weighted average of the MTP losses across all depths $\mathcal{L}_{\text{MTP}} = \lambda \sum_{k=0}^{D} \mathcal{L}_{\text{MTP}}^{k}$. Therefore, the MTP heads can be computed in an arbitrary order, and each output head only depends on the output of corresponding MTP transformer block or the backbone. *2) One MTP head can be further partitioned.* Each MTP loss $\mathcal{L}_{\text{MTP}}^{k}$ is the averaged cross-entropy of all tokens. Therefore, one output head computation can be inherently partitioned along the sequence dimension, and the computations of tokens (or token segments) are mutually independent. We can decouple the computation of the output head from the transformer. We can treat the MTP transformer as the transformer blocks in the backbone, and treat the MTP head as the output head in the backbone. Therefore, we can shard each output head into multiple sub-tasks, and execute them across different pipeline stages. Finally, we can recover results equivalent to the original computation via lightweight loss aggregation at the pipeline tail.

In this paper, we propose **AdaHC**, an adaptive pipeline scheduling framework for LLMs with MTP blocks. AdaHC performs **Head Chunking** to split and reassemble the out-

put heads into a set of normalized chunks whose cost is comparable to that of a transformer layer, and these chunks are uniformly represented by a **Chunk Descriptor Table (CDT)**. A hierarchical CDT **Planner** then maps the normalized chunks to pipeline stages, reducing schedule bubbles while eliminating imbalance bubbles. Finally, a CDT **Executor** manages inter-stage dataflow via adaptive activation forwarding and low-overhead loss aggregation, avoiding high-frequency synchronization while preserving numerical equivalence. Therefore, AdaHC fundamentally removes heterogeneity caused by multiple output heads and enables efficient, well-balanced pipeline training (an example in Figure 1(b)). Specifically, the key contributions are as follows:

- We identify the parallel nature of the MTP blocks in LLMs, including the independency among output heads and the independency among tokens in each output head, which provides a new perspective for efficient pipeline scheduling with MTP blocks.

- We introduce AdaHC, a new pipeline scheduling framework for MTP. Through Head Chunking and the CDT abstraction, it unifies unbalanced output heads and transformer layers into load-balanced scheduling units, fundamentally addressing pipeline bubble issues.

- We conduct extensive experiments across multiple LLMs with MTP. Experimental results show that AdaHC significantly improves training throughput while exhibiting strong scalability and generality.

**Conflict of Interest Disclosure.** Authors YW, CS, ZZ, MY, and YL are employed by Alibaba Group, which leads the development of Qwen, one of the models evaluated in this paper.

## 2. Background

### 2.1. Multiple Token Prediction in LLMs

The standard training paradigm for contemporary LLMs is autoregressive modeling based on *Next-Token Prediction*

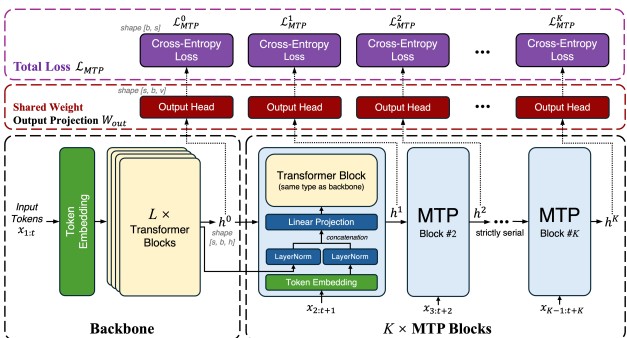

*Figure 2.* Typical model architecture of LLM with MTP.

(NTP) (Achiam et al., 2023). Given a context of length $t$, the model predicts the conditional probability distribution of the next token. Architecturally, this is typically implemented by appending a single output head at the end of a transformer backbone to project hidden states into the vocabulary space. While NTP is simple and stable, it faces inherent challenges in modeling long-range dependencies and suffers from limited inference efficiency (Qi et al., 2020). To circumvent these limitations, the *Multi-token Prediction* (MTP) has gained traction and has been widely adopted in state-of-the-art models such as DeepSeek-v3 and Qwen3-Next.

Figure 2 illustrates the computation flow of a typical MTP LLM. The model consists of an LLM backbone and $K(K \geq 1)$ serial MTP blocks. Note that the last transformer layer and output head in the backbone can be regarded as a special MTP block. Input tokens first go through the backbone embedding layer and $L$ Transformer blocks to produce hidden states, which are fed into the backbone output head to predict the next token. The hidden states are the passed to the first MTP block. Each MTP block is a sequentially stacked subnetwork including ① a transformer block (of the same type as in the backbone), ② an output head (shared among backbone and MTPs), and ③ other auxiliary parts (including a token embedding, two layer normalization, a linear projection, whose execution time is negligible compared to ① and ②). The hidden state produced by the $i$-th MTP block serves as the input to the $i$-th output head and the $(i+1)$-th MTP block, implying strictly serial execution across blocks. Each output head maps `hidden_states` $\in \mathbb{R}^{[s,b,h]}$ to `logits` $\in \mathbb{R}^{[s,b,v]}$, followed by a softmax over the vocabulary dimension $v$ to compute the loss. Finally, the losses from the backbone output head and all MTP output heads are aggregated at the end to form the final loss for back-propagation. In the remainder of the paper, we use the term *output head* to refer collectively to the output projection together with its associated loss computation.

## 2.2. Pipeline Parallelism

Pipeline parallelism (PP) is a core technique for training ultra-large models. It partitions model layers across multiple

devices and enables micro-batch-level concurrent execution. In standard PP, the model is divided into multiple *stages*, and each device stores and computes one or more stages. By pipelining forward and backward passes across devices, PP can effectively hide point-to-point communication overhead and substantially reduce per-device memory footprint.

Due to inherent sequential dependencies, PP inevitably introduces *pipeline bubbles* (i.e., device idle time). Pipeline bubbles contain three parts. Specifically, *1) Schedule bubble* (gray blocks in Figure 1) arises from pipeline warmup phase and cooldown phase, due to the limited partition granularity. Interleaved pipeline (Narayanan et al., 2021) further split each stage into several sub-stages to reduce the schedule bubble. But when a stage cannot be split further, even aggressive pipeline schedules cannot significantly reduce this bubble. *2) Imbalance bubble* (black blocks in Figure 1) arises from inter-stage workload imbalance, because the pipeline throughput is bounded by the slowest stage, while other stages stall in steady phase. Modern pipeline schemes such as 1F1B (Narayanan et al., 2019), Interleave (Narayanan et al., 2021), and DualPipe (Liu et al., 2024) primarily focus on reducing schedule bubbles, typically assuming that stages are already load-balanced. *3) Data bubble* arises from the imbalanced micro batches, such as variable sequence lengths (Yuan et al., 2025; Bai et al., 2024). In this paper, we focus on the schedule bubble and the imbalance bubble, which are related to model architectures (i.e., with MTP blocks).

## 3. Observations

Based on a detailed analysis of the computation flow in MTP architectures, we distill two key observations that provide the theoretical basis for a new pipeline scheduling scheme, which breaks the conventional view of treating an MTP head as a monolithic operator.

**Observation 1: MTP heads can be executed in out-of-order manner, due to no data dependence across them.** Unlike the strictly sequential dependencies across backbone layers (where layer $l+1$ must wait for the output of layer $l$), the dependency between output heads and the transformer backbone is inherently *loosely coupled*. Specifically, each output head depends exclusively on its input hidden states (typically the output $h_L$ from a specific backbone or transformer layer of its MTP block). Once these hidden states are computed and accessible on a given device, the corresponding MTP head can be computed independently, without further dependence on the subsequent layers of the backbone or the execution order of output heads.

Therefore, we can leverage this independence for the parallel computing of output heads. From a *spatial partitioning* perspective, heads need not be rigidly placed at the tail of

the last PP stage. As long as the required hidden states can be accessed at low cost and the resulting loss (or its contribution) can be correctly propagated to the last stage, head computation can be migrated to earlier stages or distributed across stages. From a *temporal scheduling* perspective, heads can be treated as *deferrable computational tasks*. Instead of being bound to a fixed execution slot, they can be flexibly inserted into any available idle periods in the pipeline timeline after their prerequisite hidden states have been generated. Overall, heads are not intrinsically bound to a fixed tail stage; they are better viewed as a set of flexibly placeable computational tasks. By orchestrating the input hidden states and output partial loss, we can significantly relax the spatial and temporal constraints on head execution while maintaining numerical correctness.

**Observation 2: Partitioning the output head along the sequence dimension avoids `softmax`-induced synchronization.** From the parallel computing perspective, `softmax` makes vocabulary dimension ($v$) partitioning communication-heavy due to its global normalization. As shown in Equation(1), each shard result must be corrected using the global $max$ and $sum$ (typically via cross-stage all-reduce), which causes frequent synchronization. Here, $z$ denotes the `softmax` operation, $Y_i$ denotes the $i$-th token, and $K$ denotes the number of partition participants.

$$z(Y_i) = \sum_{i=1}^{K} z'(Y_i^v) \times \frac{sum_i' \times e^{max_i' - global(max_i)}}{global(sum_i)}$$

(1)

In contrast, partitioning along the sequence dimension ($s$) is naturally computation independent: each sub-sequence still performs a *full* vocabulary softmax for its own token subset, requiring no cross-stage synchronization. The final result is obtained by simply aggregating the shards as $z(Y_i) = \sum_{i=1}^{K} z'(Y_i^s)$, without any global synchronization.

From the memory perspective, $s$-dim partitioning requires each participating device to hold a replica of the output-head weights, whereas $v$-dim partitioning only stores the corresponding weight slice. Assuming an output head is partitioned across $K$ pipeline stages, $s$-dim partitioning introduces $K-1$ additional replicas of the output-head weights compared to $v$-dim partitioning, while the activation memory footprint remains the same. On the other hand, since the backbone head shares parameters with multiple MTP heads, these replicas can be reused across heads, amortizing the extra replication cost of $s$-dim partitioning. Moreover, later pipeline stages often have more available memory due to a smaller activation footprint. Therefore, scheduling the output head with $s$-dim partitioning only on later pipeline stages does not change the global peak training memory. Consequently, trading a modest amount of memory redundancy for a synchronization-free execution pattern provides

*Table 1.* Terminology and explanations.

| Terms | Explanation |
|---|---|
| Task | Specific computational tasks (eg. a TF block or an OH chunk). |
| Stage | The model partition residing on a device. It contains several slots in AdaHC. |
| Slot | A logical scheduling position within a stage for placing fixed-cost tasks. |
| Entry | A record in the CDT that describes one task and its schedule. |

superior scalability for pipeline scheduling.

## 4. Design of AdaHC

To address the *schedule* and *imbalance* bubbles introduced by MTP architectures under PP, we propose **AdaHC**, an adaptive pipeline scheduling framework tailored to MTP LLMs. **The key idea is to reshape output heads into flexible sub-tasks that adapt to pipeline parallelism.** AdaHC can simultaneously mitigate limited scheduling granularity (to reduce schedule bubble) and inter-stage workload imbalance (to reduce imbalance bubble), while keeping low communication overhead. Figure 3 overviews AdaHC's design. AdaHC consists of three modules: (1) **CDT constructor**: Construct a *Chunk Descriptor Table* (CDT) that uniformly represents transformer layers and output head chunks as scheduling tasks; (2) **CDT Planner**: Generate a pipeline scheduling plan following the data dependency between transformer layers and output heads. Specifically, it maps CDT entries to specific PP stages and determines the inter-stage execution order, and applies intra-stage operator-fusion optimizations; and (3) **CDT Executor**: Execute the scheduling plan and manages activation forwarding and loss aggregation for MTP heads, ensuring numerical equivalence with controlled communication overhead.

### 4.1. Head Chunking and CDT Constructor

Chunk Descriptor Table (CDT) uniformly represents transformer layers and output head chunks as scheduling entries The goal is to reconstruct both transformer computation and output head computation into a set of basic scheduling units with aligned compute cost, providing a foundation for subsequent pipeline mapping and load balancing.

According to Section 3, MTP head computation can be losslessly partitioned along the sequence dimension ($s$). We decompose all MTP head computation into a set of *head chunks*, where each chunk is defined as Equation(2). The CDT uniformly represents transformer layers and MTP head chunks as scheduling entries. The goal of CDT is to decomposing the LLM model into a set of basic scheduling units with aligned computation cost, providing a foundation for further pipeline mapping and load balancing.

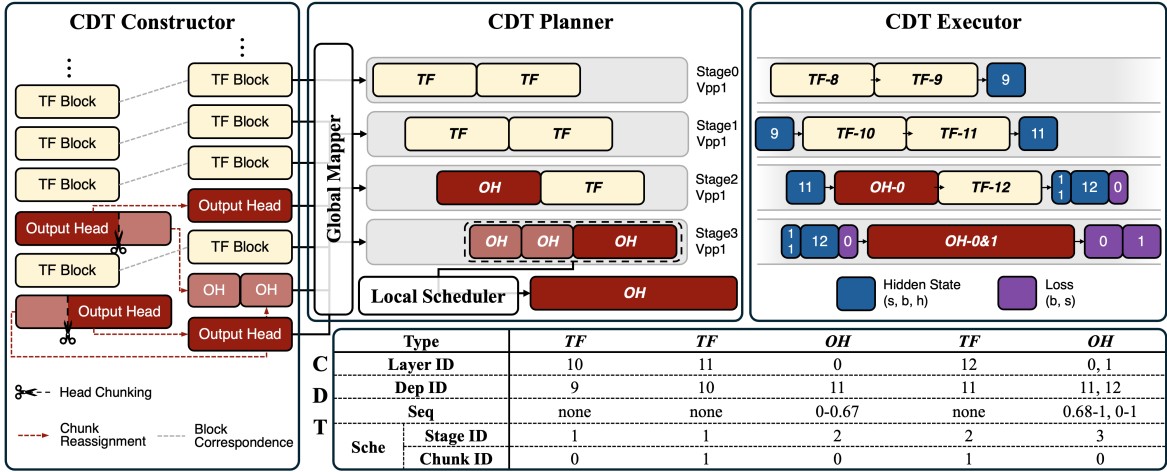

*Figure 3.* Design overview of AdaHC. It contains three modules: CDT constructor, CDT planner, and CDT Executor.

$$HeadChunk = \{(k_i, [s_{a,i}, s_{b,i}]) \mid i = 1, 2, \ldots, N\},$$
$$S_{\text{headchunk}} = \sum_i (s_{b,i} - s_{a,i}), \quad (2)$$

where $(k_i, [s_{a,i}, s_{b,i}])$ denotes the sequence interval $[s_a, s_b]$ for the $k$-th MTP head. To achieve precise workload alignment, we introduce a lightweight cost model to determine the optimal chunk size, as shown in Equation(3).

$$S_{\text{headchunk}} = \text{solve} \quad s \mid T_{OH}(s) \approx T_{TF} \quad (3)$$

where $T_{TF}$ represents the empirically measured execution latency of a single transformer layer, and $T_{OH}(s)$ denotes the estimated latency for an output head with sequence length $s$, derived from its FLOPs and kernel profiling. Figure 3(a) illustrates the sequence-wise partitioning of two output heads into standardized head chunks. Appendix B.2 further analyzes the impact of head chunk granularity on training throughput.

AdaHC abstracts all computational tasks, including transformer layers (exist both in backbone and in MTP blocks) and the newly generated head chunks, into entries in the **CDT**. AdaHC adopts the CDT as the core abstraction for the pipeline scheduling, with each entry containing the following metadata:

- **Type**: Identify the unit type, e.g., a transformer layer (TF) or an output head (OH).
- **ID**: Layer index and the dependent layer index.
- **Seq**: The sequence span covered by OH, e.g., $[0, 0.67]$.
- **Schedule (Sche)**: Scheduling fields assigned by the CDT Planner including pipeline stage and chunk index within the stage.

This CDT abstraction offers two-fold benefits. *1)* With CDT, the monolithic head computations, which are typically clus-

---

**Algorithm 1** CDT Planner

**Input:** CDT, PP_schemes (e.g., 1F1B, Interleave)
**Output:** updated CDT, $pp\_schedule$
sort(CDT, key ← lambda x: (x.Dep ID, x.Type != 'OH'))
$pp\_schedule$ ← generate_schedule(PP_schemes, len(CDT))
**# Global Mapper (greedy)**
**for each** $(entry, layer)$ **in** $(CDT, pp\_schedule)$ **do**
    $s_0$ ← GetCurrentStage(entry.dependent_layer)
    $s$ ← $\big($HasSlot($pp\_schedule[s_0]$) ? $s_0$ :
        NextStageWithSlot($pp\_schedule, s_0$)$\big)$
    entry.stage_id ← $s$
    entry.chunk_id ← appendInOrder($pp\_schedule[s]$, entry)
**end for**
**# Local Scheduler (head chunk fusion)**
**for each** stage **in** $pp\_schedule$ **do**
    local_head_chunks ← get_all_head_chunks_in(stage)
    **if** count(local_head_chunks) > 1 **then**
        fused_chunk ← fuseHeadChunks(local_head_chunks)
    **end if**
    update_schedule(CDT, stage, fused_chunk)
**end for**
**return** CDT, $pp\_schedule$

---

tered at the final pipeline stage, can be decomposed into multiple CDT entries with the same granularity as transformer layers. It allows enabling the pipeline schemes with finer stage granularity (e.g., Interleave), thereby reducing the *schedule bubble*. *2)* With CDT, we can ensure that the cost of each scheduling unit is approximately uniform, which helps alleviating pipeline scheduling overhead. So that the CDT planner can easily achieve inter-stage load balancing to reduce the *imbalance bubble*, and focuses more on further performance optimizations.

## 4.2. CDT Planner

AdaHC employs a hierarchical **CDT Planner** to map scheduling units onto physical devices. As depicted in Figure 3(b), the planner consists of two components: a *Global*

*Mapper* for high-level task allocation and a *Local Scheduler* for low-level operator optimization. The core scheduling logic is detailed in Algorithm 1.

**Dependency-Aware Global Mapper: Map CDT entries to devices with minimized inter-device communication.** The global mapper first performs a topological sort on all CDT entries based on their data dependencies ($CDT.dep$). It adopts a "*Prioritize Output Head (OH)*" heuristic for entries with identical dependency levels. This heuristic ensures that output heads are placed as close as possible to their prerequisite transformer layers, eliminating unnecessary cross-device communication of hidden states.

Then, the global mapper initializes a set of logical scheduling slots $\mathcal{P}$ on each pipeline stage, and the slot number is determined by the given pipeline schemes (e.g., 1F1B or Interleave), and greedily assigns CDT entries to scheduling slots. For each OH entry, the global mapper first tries mapping to slots in the same stage as its dependent transformer layer. If the all slots in stage are occupied, it searches for the nearest downstream stage with available slots. Only when an OH entry packs multiple distinct output heads, mapping it to a downstream stage incurs additional hidden-state forwarding. However, the overhead is orders of magnitude lower than the hardware idle time caused by the imbalance bubble. TF entries are mapped in dependency order, and when a conflict arises with an OH entry's mapping, the OH entry is prioritized. In practice, this greedy mapping avoids unnecessary hidden-state forwarding. Upon completion, each CDT entry is assigned a precise physical stage index ($stage\_id$) and a local execution order ($chunk\_id$).

**GEMM-Aware Local Scheduler: Fuse intra-stage head chunks.** The local scheduler inspects each stage to identify multiple independent head chunks. Since kernels on small chunks often underutilize modern GPUs, the local scheduler tries the best to fuse them. Without violating CDT dependencies, it merges head chunks assigned to the same stage along the sequence dimension into a larger GEMM kernel, enhancing computational intensity for better GPU utilization and amortizes kernel launching overhead. The $Sche$ of CDT is updated accordingly to reflect these fusions. Note that even these chunks are from different MTP blocks, the local scheduler can still fuse them. Because the output heads in a LLM share the same parameters, the local scheduler only needs to correctly concatenate and split the input and output tensors along the sequence dimension.

### 4.3. CDT Executor

After the CDT planner generates and back-fills the complete $Sche$ metadata, **CDT Executor** materializes the plan and manages runtime dataflow in the pipeline. This design preserves the inherent sequential dependencies among transformer blocks. Furthermore, it ensures that each *head chunk*

can access its required input hidden states and that per-chunk loss values are aggregated to the final stage with minimal overhead for global backpropagation. In Figure 3(c), purple blocks denote cross-stage hidden states tensors (shape $[s, b, h]$), and red blocks denote the loss tensors (shape $[b, s]$) computed by head chunks and incrementally aggregated along the pipeline. The dataflow management for head chunks is implemented via the following two mechanisms.

**Adaptive Activation Piggybacking.** Depending on the global mapping, an OH and its dependent TF or the next TF may be co-located or split across stages, and the CDT executor handles both cases adaptively. If an OH is co-located with its dependent TF, the executor directly reads the corresponding hidden states from local activations. If a head chunk is mapped to a later stage, the executor enables a *piggybacking* forwarding strategy: once the required hidden states are produced, the executor appends the activation tensor $\mathcal{A}$ (the hidden states consumed by that OH chunk) to the mandatory pipeline payload $\mathcal{C}_f$, and forms the $\mathcal{C}_f|\mathcal{A}$ tensor. Tensor $\mathcal{A}$ is then carried along the pipeline hop by hop until it reaches the destination stage that hosts the head chunk. Upon receiving $\mathcal{C}_f|\mathcal{A}$, the executor extracts the appended tensor $\mathcal{A}$ and feeds it to the corresponding head chunk computation. If a stage contains only OH but no TF, the executor continues forwarding $\mathcal{C}_f$ and remaining $\mathcal{A}$ that has not yet been consumed. All such forwarding is strictly on-demand, triggered only when cross-stage transmission is necessary, minimizing additional communication.

**Bit-wise Transparent Loss Aggregation.** Each head chunk produces a local loss tensor, typically of shape $[b, s]$. Compared to the hidden states $[s, b, h]$, the loss is small but often uses higher precision (Achiam et al., 2023). To preserve the numerical precision while avoiding frequent small-message communications in the multiple MTP setting, the CDT Executor employs a *bit-level transparent tensor packing* approach. Without changing the underlying bit pattern, it reinterprets the loss tensor as a lower-precision tensor $\mathcal{L}$ with multiples of original size, and appends it to the tail of $\mathcal{C}_f$, which forms $\mathcal{C}_f|\mathcal{L}$. For example, when the hidden states are in bfloat16 format and the loss is in float32 format, the loss is packed into a tensor of shape $[s, b, 2]$ and appended after the hidden states, resulting in a combined tensor of shape $[s, b, h + 2]$ for communication. Upon receiving this tensor, the executor performs the inverse reinterpretation after extraction, enabling loss transmission without precision loss. Our internal analysis shows that the induced communication overhead is only a $\frac{\eta}{h}$ fraction of the primary payload, where $\eta$ is the bit-width ratio between loss and hidden states, and $h$ is the hidden-state dimension. In practice, this overhead is negligible, as detailed quantified in Appendix B.4.

*Table 2.* Model configurations. To emulate compute-intensive MTP settings, we equip Qwen3-30B and Qwen3-235B with the same MTP architecture as Qwen3-Next-80B.

| Model | Num of Layers | Hidden Size | FFN Size | Top-K | MTP |
|---|---|---|---|---|---|
| Qwen3-8B | 36 | 4096 | 12888 | – | – |
| Qwen3-30B-A3B | 48 | 2048 | 768 | 8 | 1 |
| Qwen3-235B-A22B | 94 | 4096 | 1536 | 8 | 1 |
| Qwen3-Next-80B-A3B | 48 | 2048 | 512 | 10 | 1 |

# 5. Evaluation

## 5.1. Experimental Setup

We implement AdaHC on top of the Megatron-LM framework and run all experiments on an NVIDIA GPU cluster. Intra-node GPUs are connected via NVLink, while inter-node communication uses RoCE. Unless otherwise specified, all methods are evaluated under an identical training software stack and runtime configuration (eg., recomputation granularity and the Transformer Engine version v2.0.0), to ensure that throughput differences are solely attributed to pipeline partitioning and output head scheduling strategies.

**Models.** To validate the scalability of AdaHC in production-like settings, we evaluate a range of open source models in Table 2. For Qwen3-Next-80B, to match cluster resources, we reduce the number of MoE experts to 128 while keeping the top-$k$ routing hyper-parameters unchanged.

**Baselines.** We compare AdaHC with three PP partitioning strategies under 1F1B or interleaved 1F1B. For each baseline, we tune parallelism to the best of our efforts under identical system constraints. Detailed parallelism configurations are provided in Appendix D.

- **Naive Baseline (`Naive`):** Adopts a uniform transformer layer distribution and places all MTP heads in the final pipeline stage (Megatron-LM default), which often leads to a tail-heavy *imbalance bubble*.
- **Vocabulary Parallel (`VP`):** Shards the output head along the vocabulary dimension within each pipeline group to reduce per-GPU head compute, at the cost of extra communication and synchronization. We adopt its open-source implementation[1], which supports only the single MTP settings, thus only reported in single-MTP experiments.
- **Layer Redistribute (`LR`):** Moves several transformer layers from the final stage to earlier stages to hide the computational load of MTP heads and mitigate load imbalance. We select the optimal LR configuration within GPU memory capacity

**Metrics.** We use the throughput (**tokens per GPU per second**) as the primary metric for training efficiency. To facilitate comparison across model scales, we normalize throughput within each experimental group (by default, rel-

---

[1] https://github.com/sail-sg/VocabularyParallelism

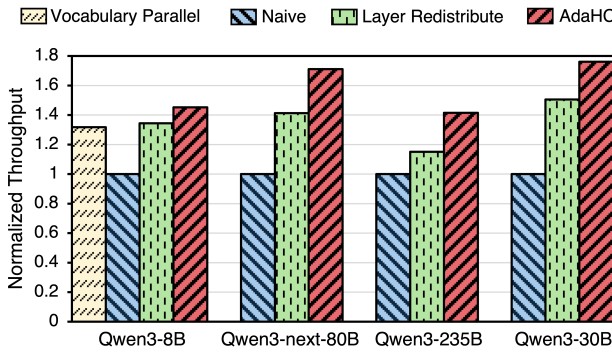

*Figure 4.* End-to-end throughput among four PP schemes under different model configurations.

ative to the `Naive` in that group).

## 5.2. End-to-end Throughput

We compare the end-to-end training throughput of AdaHC against the baselines on four models in Table 2. Figure 4 shows AdaHC consistently delivers substantial speedups across all evaluated settings.

**No MTP case (only an output head in backbone).** On Qwen3-8B, we compare AdaHC with `Naive`, `LR`, and `VP`. AdaHC improves throughput by 45.24% over `Naive`. While `VP` and `LR` partially alleviate the tail-stage compute pressure via head parallelization or heuristic layer re-balancing, AdaHC still outperforms them by 10.29% and 8.09%, respectively. This indicates that even in the no MTP scenario, AdaHC's fine-grained head chunking achieves more precise load balancing than `LR`, while avoiding the frequent cross-device synchronization introduced by `VP`.

**One MTP case (#MTP=1).** For the Qwen3-30B, Qwen3-Next-80B, and Qwen3-235B, we compare AdaHC with `Naive` and `LR`. AdaHC improves throughput by 76.05%, 71.11%, and 41.46% compared to `Naive`, and by 16.96%, 21.11%, and 22.93% compared to `LR`. With MTP, the tail stage becomes substantially heavier, exacerbating inter-stage imbalance and enlarging the imbalance bubble, making AdaHC more beneficial. Although `LR` can move more transformer layers toward earlier stages to cover the tail overhead, the coarse-grained layer partition and GPU memory budget prevent precise stage-level load balance. In contrast, AdaHC decomposes the output heads into independently schedulable head chunks, balancing stage-wise computation and yielding consistently superior end-to-end throughput.

## 5.3. Load Balance Analysis

To attribute the throughput gains, we profile the per-stage forward time (including both computation and communication) for the experiments in Section 5.2. Figure 5 reports the CDF of stage-time imbalance, where the x-axis is the threshold $t$ and the y-axis is the fraction of GPUs satis-

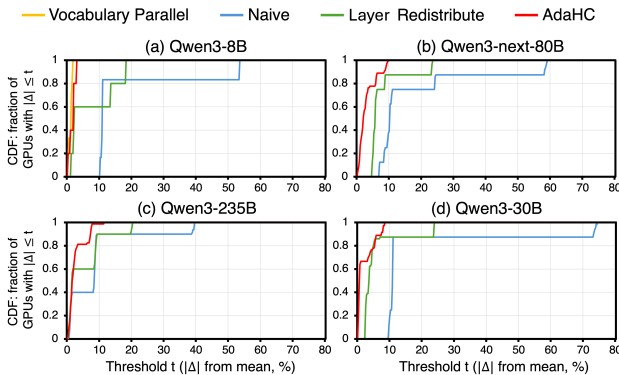

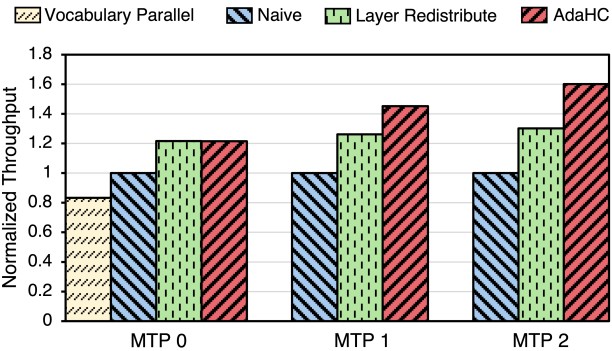

*Figure 5.* Load balance of `Naive`, VP, LR, and AdaHC. The x-axis is the imbalance threshold $t$ (percentage absolute deviation $|\Delta|$ from the mean GPU load), and the y-axis is the CDF (Cumulative Distribution Function), the fraction of GPUs with $|\Delta| \leq t$.

*Figure 6.* Throughput under varying numbers of MTP blocks.

fying $|\Delta| \leq t$. Curves that rise faster (i.e., reach a high CDF at small $t$) indicate more uniform stage workloads and fewer imbalance bubbles. Across all experiments, AdaHC achieves the steepest curves, showing that head chunking yields near-uniform stage workloads. VP also achieves relatively good balance but incurs substantial communication overhead. As analysis in Appendix C shows, the additional communication time introduced by VP at each stage can even exceed the compute time of the chunked output head itself, limiting its end-to-end benefits. Moreover, LR can only migrate the whole transformer layer to compensate for the tail-stage overhead. Its adjustment granularity is limited and is often constrained by GPU memory, making it difficult to achieve well-balanced workloads.

### 5.4. Scaling MTPs

To examine how AdaHC performs as the number of MTP block scales, we conduct a controlled experiment on Qwen3-8B. Concretely, under a fixed computation and memory budget, we truncate the backbone to 29 layers, so that we can systematically vary the ratio of computation time between transformer and MTP head. We compare the throughput of AdaHC against the baselines under **no MTP**, **1 MTP block**, and **2 MTP blocks**, as shown in Figure 6.

AdaHC shows better speedup with more MTP blocks. Compared to `Naive`, AdaHC improves throughput by 21.56%, 45.15%, and 60.08%. When MTP is absent, LR is already near-optimal balance. Although AdaHC achieves finer-grained balance, the interleaved schedule reduces the total layers per stage, which hurts computational efficiency slightly for small models. However, under 1 and 2 MTP(s), LR is bounded by its adjustment granularity, trailing AdaHC by 15.09% and 22.95%, respectively. Moreover, without MTP, AdaHC improves throughput over VP by 45.30%. This is because VP incurs substantial synchronization overhead, making it even slower than `Naive` in this setting. Appendix C provides a detailed analysis.

## 6. Related Works

In LLM training, most studies on pipeline parallelism have primarily focused on reducing pipeline bubbles (Narayanan et al., 2019; Li & Hoefler, 2021; Liu et al., 2023), overlapping computation with communication (Liu et al., 2024; Qi et al., 2023), balancing memory pressure (Kim et al., 2023; Wan et al., 2025), and activation checkpointing (Liu et al., 2025a; Sun et al., 2024). Recent studies have also identified the inter-stage computational imbalance induced by output head. For example, Skywork-MoE (Wei et al., 2024) and DeepSpeed (Smith et al., 2022) try to mitigate this issue via static layer reordering. However, due to the coarse granularity of whole-layer assignment and the additional memory overhead induced by moving layers earlier in the pipeline, such reordering strategies are often less effective on multiple MTP architectures. Vocabulary parallel (Tsung et al., 2025) can effectively balance inter-stage workload in single output head cases, but it introduces substantial communication and synchronization overhead and does not readily extend to multiple output head cases. SlimPipe (Li et al., 2025) employs an approach akin to vocabulary parallel, but points out that it is not straightforward to incorporate into standard PP schemes due to their misaligned schedules. Furthermore, techniques proposed for inference, such as micro-batch size adjustment (Zhang et al., 2025; Guo et al., 2025; Agrawal et al., 2023) and dynamic layer reordering (Xu et al., 2025), are inapplicable to training.

## 7. Conclusion

With AdaHC, we successfully demonstrate the feasibility of reducing pipeline bubbles on training LLMs with MTP block(s), achieving significant training throughput improvements. By transforming computationally heavy output heads into flexible, load-balanced scheduling units through the CDT abstraction, AdaHC fundamentally addresses inter-stage load imbalance and enables finer-grained pipeline partitioning to reduce schedule bubble without significant communication and synchronization overhead. Extensive experiments across multiple Qwen3-family models

show that AdaHC consistently improves end-to-end training throughput under diverse MTP configurations and scales better as the number of MTP blocks increases. Looking forward, we hope AdaHC will motivate broader exploration of architecture-aware pipeline scheduling, paving new directions for practical and efficient pipeline training of next-generation LLMs with complex prediction architectures.

## Acknowledgments

This work is supported by the following funding: National Science Foundation of China (92270206, 62372435, 62502501), Beijing Natural Science Foundation (4254087), the Innovation Funding of ICT, CAS.

## Impact Statement

This paper presents work whose goal is to advance the field of machine learning. There are many potential societal consequences of our work, none of which we feel must be specifically highlighted here.

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

## A. Why AdaHC Works: Rescheduling Non-Blocking Computation in PP

In PP training, end-to-end throughput is ultimately governed by the critical path—the computations whose output activations must be consumed immediately by subsequent layers, thereby determining when the next stage can make progress. In contrast, training graphs often contain non-critical-path modules: their computation must finish within an iteration, but their outputs are not an immediate dependency of subsequent modules and are only consumed later (e.g., for loss aggregation at the end of the forward pass or for gradient updates in the optimizer).

AdaHC systematically identifies and reschedules such non-critical computations (e.g., on LLMs with MTP block(s)). Without changing numerical equivalence, it disperses them from hotspot stages into slack across multiple stages, improving inter-stage load balance, reducing imbalance bubbles, and enabling finer-grained pipeline schedules (e.g., interleaved 1F1B). MTP is a representative instance of such reschedulable non-critical computation, and thus serves as our primary analysis and evaluation setting; however, AdaHC is not specific to MTP.

## B. Supplementary Experiments and System Analyses of AdaHC

### B.1. Accuracy Verification

To verify that AdaHC preserves the numerical semantics of the original training task while substantially improving training efficiency, we compare its convergence behavior against `Naive` in a controlled setting. We truncate Qwen3-30B to **19 layers** and insert **one MTP module** at the end of the backbone. All other training hyperparameters and parallelism configurations are held constant (Table 6). Figure 7 plots the training loss as a function of optimization steps for AdaHC and the baseline under the same number of steps. AdaHC closely matches the baseline throughout training, with no systematic loss shift or convergence degradation.

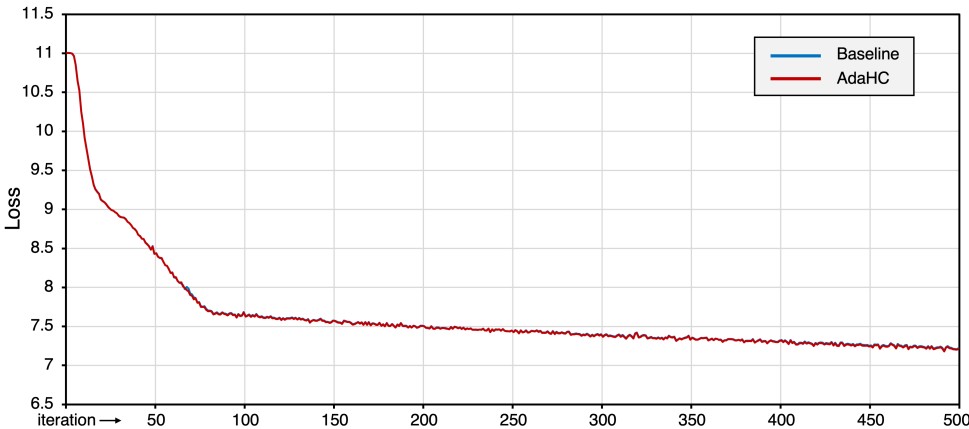

*Figure 7.* Loss curves for `Naive` baseline and AdaHC training with same configurations.

### B.2. Head Chunking Size Analysis

In the end-to-end evaluation in Section 5.2, AdaHC's speedup mainly comes from chunking the output head and remapping head chunks across pipeline stages to mitigate the tail-end compute hotspot. To better understand how chunk size affects performance, we use Qwen3-30B as a case study to examine throughput under different split ratios, and summarize takeaways that generalize to other configurations.

Under the Qwen3-30B setup, the CDT Planner distributes the output-head computation across the last two pipeline stages. We keep all other parallelization and scheduling settings fixed, and only vary the fraction of the output-head sequence assigned to the penultimate stage (i.e., the *split width*). Concretely, *split width* denotes the proportion (along the sequence dimension) of the head chunk executed on the penultimate stage, with the remaining portion executed on the final stage. Figure 8 reports the normalized throughput under different *split width* values (normalized to the worst point). We observe up to $\sim 10\%$ throughput variation: throughput peaks around *split width* $= 0.9$ and degrades when the split is either too coarse or too fine.

We attribute this behavior to how well the head-chunk size matches the per-stage workload. When the head chunk is too large (coarser split), the head-bearing stage becomes significantly slower than stages containing only Transformer layers. As a result, in steady state the pipeline step time is dominated by this slow stage, inducing larger **imbalance bubbles** across nearly all stages and causing a larger throughput drop. When the head chunk is too small (finer split), the head-bearing stage can become faster than a pure-Transformer stage; the resulting imbalance is then more localized to the few stages that host head chunks, leading to a smaller throughput impact. However, when the chunk becomes even smaller, output-head computation can accumulate on the tail stages due to limited available slots and mapping constraints, re-forming a "slow tail" hotspot; imbalance then grows again and throughput decreases. In this experiment, *split width* $= 0.95$ degrades more noticeably than $0.85$, indicating that overly fine splitting already triggers the tail-accumulation/mapping-limited effect in this configuration. As the size further deviates from the optimum, we consistently observe clear performance degradation.

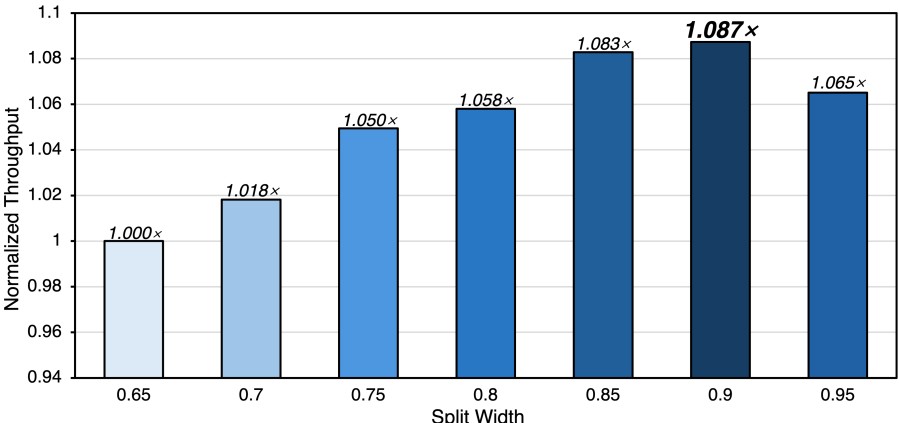

*Figure 8.* Effect of chunking size on AdaHC's throughput.

Overall, head-chunk size must balance two competing goals: (i) smoothing the tail workload, and (ii) avoiding tail-stage accumulation that creates stage-level hotspots. Overly large chunks tend to induce pipeline-wide steady-state imbalance, while overly small chunks can cause hotspots via tail accumulation. In practice, a more robust guideline is to first profile the runtime of a Transformer layer and the output head under the target configuration, and then exploit the fact that output head cost is approximately proportional to sequence width to tune the chunk size such that each head chunk's time is close to (and slightly smaller than) that of a Transformer layer, yielding more stable throughput across models and parallelization settings.

### B.3. Memory Analysis

To enable flexible chunking of the output head along the sequence dimension in AdaHC, all pipeline stages that participate in the corresponding computations maintain a replica of the output-head weights. To assess the memory overhead of this design, we measure and compare the peak GPU memory usage of AdaHC across pipeline ranks in the experiments from Section 5.2. The detailed training configuration is provided in Appendix D. As shown in Fig. 9, the pipeline exhibits a pronounced "front-heavy, back-light" memory footprint: in pipeline-parallel training, activations produced by earlier stages must be retained across more subsequent micro-batches before they are consumed by backpropagation, whereas later stages are closer to the loss, so their activations enter backpropagation sooner and are freed earlier, resulting in a lower peak memory.

We also observe a noticeable "bump" in the peak memory of the penultimate pipeline stage. This arises because AdaHC chunks the output head—which is otherwise placed entirely on the last stage—and redistributes these chunks across the last few stages to balance computation. Since the output head introduces additional static parameter storage and dynamic activations, it can create local peak memory increases. However, this bump is typically not the global peak and the activation component can be mitigated via recomputation. An exception occurs in Fig. 9(c) for Qwen3-235B, where the penultimate stage becomes the global peak. This is because training enables full recomputation for LayerNorm, Attention, and MoE, replacing the storage of major activations with recomputation, while the output-head activations are not recomputed, making them dominate the global peak memory.

Overall, the weight replication introduced by AdaHC primarily affects the tail stages of the pipeline. Since the pipeline backend naturally sits in a memory valley, and AdaHC's flexible chunk scheduling places the additional memory pressure

from the output head onto these memory-abundant tail stages, the design exploits the intrinsic imbalance of memory usage across pipeline stages. Consequently, AdaHC achieves substantial throughput gains by trading a fraction of redundant backend memory capacity, without increasing the global peak memory.

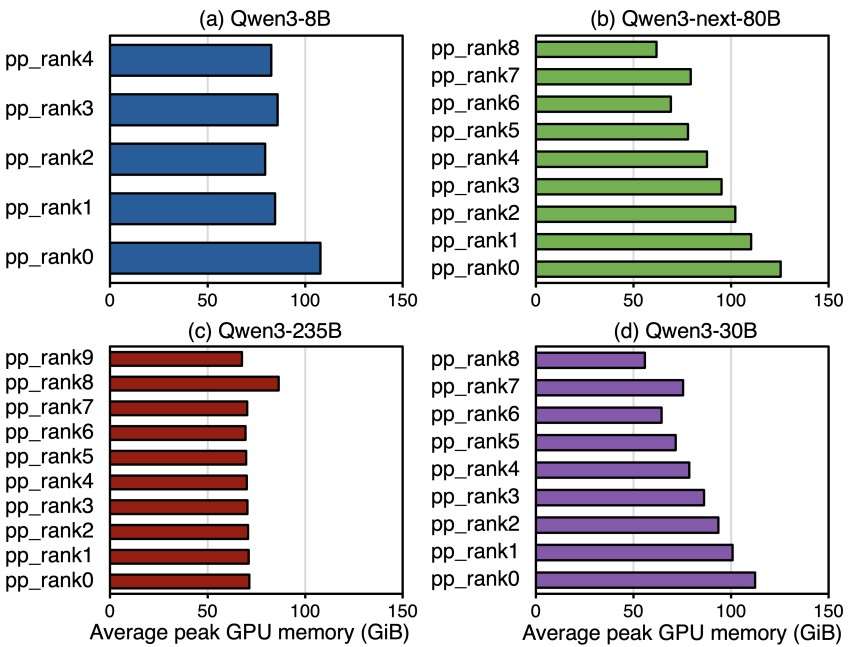

*Figure 9.* Peak GPU memory footprint of AdaHC across different pipeline stages. Note that Qwen3-235B enables recomputation for layer normalization, attention, and MoE, which eliminates most activations and yields a relatively balanced memory footprint across pipeline stages. As a result, the second last stage without output-head recomputation becomes the peak-memory stage. **We enabled recomputation only after observing OOM on the first pipeline stage during training.** Detail configurationss are in Appendix D.

### B.4. Piggybacking Pipeline Communication Overhead

A key design in AdaHC is a *piggybacking* mechanism that resolves cross-stage data dependencies without introducing extra standalone communication operators, thereby preserving efficient distributed execution. Concretely, during pipeline-parallel execution, AdaHC attaches the inputs required by head chunks (hidden states) or the computed partial losses to the standard pipeline point-to-point (P2P) messages. That is, we extend the payload on the existing activation send/recv path rather than issuing additional explicit send/recv calls.

*Table 3.* Communication overhead analysis.

|  | Shape | Dtype | Size | Theoretical communication time |
|---|---|---|---|---|
| Qwen3-8B | [4096, 2, 2] | bf16 | 0.033MB | $6.6 \times 10^{-3}$ us |
| Qwen3-next-80B | [4096, 2, 2048+2] | bf16 | 33.59MB | 671 us |
| Qwen3-235B | [4096, 1, 4096+2] | bf16 | 33.57MB | 671 us |
| Qwen3-30B | [4096, 2, 2048+2] | bf16 | 33.59MB | 671 us |

Table 3 reports the maximum additional payload and its theoretical latency for the model settings in Section 5.2. Qwen3-8B only requires piggybacking the loss, while the other three models require piggybacking both the loss and hidden states; in all cases, the corresponding communication time is at the microsecond scale. More importantly, this extra traffic follows the same P2P path as native pipeline communication and introduces no additional all-reduce/barrier-style synchronization. Under 1F1B or interleaved 1F1B, pipeline P2P transfers typically overlap with computation from neighboring micro-batches; as a result, although piggybacking increases the message size, its *visible* end-to-end overhead is often far below the theoretical upper bound and rarely becomes a throughput bottleneck.

## C. Quantitative Analysis of Vocabulary Parallel

This section quantitatively analyzes the benefits and costs of Vocabulary Parallelism under PP training, and explains why `VP` can improve per-stage compute balance yet often delivers limited end-to-end throughput gains. We compare the forward end-to-end latency of `Naive` and `VP` across pipeline stages in the settings of Section 5.2 and Section 5.4. Results are summarized in Table 4 and Table 5.

*Table 4.* Forward-time breakdown of `Naive` and `VP` across pipeline stages for Qwen3-8B in Section 5.2.

| $T$ | Per transformer layer time | $O$ | Per output head time |
|---|---|---|---|
| $S$ | Per splited head time | $E$ | Extra comm time |

| | Forward Time (Per PP Rank) / ms | $T$ | $O$ | $S$ | $E$ |
|---|---|---|---|---|---|
| `Naive` | (3406.73, 3383.06, 3386.83, 3383.85, 3377.09, 5811.88) | 563.71 | 2429.62 | 404.93 | 1125.94 |
| `VP` | (4984.13, 4861.16, 4989.07, 4854.69, 4902.37, 4889.16) | | | | |

*Table 5.* Forward-time breakdown of `Naive` and `VP` across pipeline stages for no MTP in Section 5.4.

| $T$ | Per transformer layer time | $O$ | Per output head time |
|---|---|---|---|
| $S$ | Per splited head time | $E$ | Extra comm time |

| | Forward Time (Per PP Rank) / ms | $T$ | $O$ | $S$ | $E$ |
|---|---|---|---|---|---|
| `Naive` | (1704.19, 1706.75, 1711.89, 1722.12, 1712.06, 2803.25) | 342.20 | 1434.45 | 239.07 | 1016.35 |
| `VP` | (2885.85, 2854.59, 2871.52, 2868.99, 3270.12, 2707.29) | | | | |

In the tables, **Forward Time (per PP Rank)** reports the total forward time observed on each PP rank, including both computation and communication. **T**, **O**, and **S** capture computation-only costs: the per-layer transformer time, the time of a full output head, and the idealized output-head time apportioned to each stage under perfect partitioning, respectively. Finally, **E** estimates the additional communication overhead by subtracting the expected compute cost from the measured total time, serving as a proxy for the extra communication and synchronization introduced by `VP`.

Table 4 and Table 5 suggest two key observations. First, `VP` indeed makes the per-stage forward time more uniform, indicating improved timing balance. However, part of this "balance" comes from communication and synchronization triggered after each stage completes its computation, which forces stages to align on the timeline; thus, improved timing balance alone does not imply higher throughput. Second, `VP` introduces substantial extra communication overhead. Although partitioning the output head along the vocabulary dimension reduces the theoretical head compute assigned to each stage (**S**), the measured extra communication time **E** can be several times larger than the apportioned head compute time. This indicates that the gains from splitting head computation are easily offset by collective communication and synchronization (e.g., cross-shard normalization for softmax and gradient aggregation), thereby limiting end-to-end throughput improvements.

Under the configuration of Section 5.4, `VP` is even slower than `Naive`. This is because `VP`'s **E** outweighs the benefits of head partitioning, while `Naive` partially mitigates the head-induced imbalance bubble by placing only four layers on the last stage. Overall, `VP` is most suitable when the number of layers per pipeline stage is already balanced and per-stage transformer compute dominates the output head. Moreover, `VP` is often difficult to directly integrate into general PP schedules, since its synchronization points are not aligned with the pipeline timeline, which can introduce communication interference and additional bubbles (Li et al., 2025).

## D. Detailed Parallel Configurations

This section provides the detailed parallelism configurations used in the end-to-end throughput experiments (Section 5.2), scaling MTPs experiments(Section 5.4) and accuracy verification experiments(Section B.1). All parallelism strategies reported here have already been extensively tuned to the best of our effort to ensure a fair and strong comparison. Table 6 refers to Qwen3-30B-L19 used in the accuracy verification experiments, and Table 7 refers to Qwen3-8B-L29 used in the MTP scaling experiments. Here, `Lxx` indicates that the number of layers of the model is modified compared to its open-source counterpart, while all other configurations remain unchanged. Tables 8, 9, 10, and 11 correspond to Qwen3-8B, Qwen3-30B, Qwen3-Next-80B, and Qwen3-235B used in the end-to-end throughput experiments, respectively. We summarize the notation below:

- **Pipeline Placement.** This column specifies the distribution of Transformer layers across pipeline stages. The number of columns corresponds to the pipeline parallel size (PP size), and the number of rows corresponds to the virtual

pipeline parallel size (virtual PP size, used by interleaved 1F1B). Each entry indicates the number of Transformer layers assigned to that stage, excluding the Transformer layers in the MTP blocks.

- **Recompute.** This column lists the modules for which recomputation is enabled. Recompute avoids storing the activations of these modules by re-running their forward pass during backpropagation. We enabled recomputation only after observing OOM on the first pipeline stage during training, and we prioritize low-overhead recomputation (e.g., layer normalization).

- **OOM (Out of Memory).** Configurations marked as OOM cannot be launched on our hardware due to exceeding GPU memory capacity.

- **Other parameters.** These include micro-batch size (BS), global batch size (GBS), data parallel size (DP), and expert parallel size (EP) (Rajbhandari et al., 2022).

*Table 6.* Accuracy setup for Qwen3-30B-L19.

| Qwen3-30B-L19 | Pipeline Placement | BS | GBS | DP | EP |
|---|---|---|---|---|---|
| Baseline | 2 3 3 3
3 3 2 0 | 1 | 180 | 2 | 2 |
| AdaHC | 2 3 3 3
3 3 2 0 | 1 | 180 | 2 | 2 |

*Table 7.* Qwen3-8B-L29 with no MTP training configurations.

| | Pipeline Placement | BS | GBS | DP | Recompute |
|---|---|---|---|---|---|
| Naive | 5 5 5 5 5 4 | 1 | 108 | 1 | none |
| VP | 5 5 5 5 5 4 | 1 | 108 | 1 | none |
| LR | 4 4 5 5 5 5 1 | 1 | 112 | 1 | none |
| AdaHC | 2 2 2 2 2 2 2 2
2 2 2 2 2 2 1 0 | 1 | 112 | 1 | none |

*Table 8.* Qwen3-8B training configurations.

| Qwen3-8B | Pipeline Placement | BS | GBS | DP | Recompute |
|---|---|---|---|---|---|
| Naive | 6 6 6 6 6 6 | 2 | 180 | 1 | layernorm |
| VP | 6 6 6 6 6 6 | 2 | 180 | 1 | layernorm |
| LR | 3 4 4 4 4
4 4 4 4 1 | 2 | 180 | 1 | layernorm |
| AdaHC | 2 2 2 2 2
2 2 2 2 2
2 2 2 2 2
2 2 2 0 0 | 2 | 180 | 1 | layernorm |

*Table 9.* Qwen3-30B training configurations.

| Qwen3-30B | Pipeline Placement | BS | GBS | DP | EP | Recompute |
|---|---|---|---|---|---|---|
| Naive | 6 6 6 6 6 6 6 6 | 2 | 1152 | 8 | 8 | none |
| LR | 5 7 7 7 7 7 7 1 | 2 | 1152 | 8 | 8 | none |
| AdaHC | 3 3 3 3 3 3 3 3
3 3 3 3 3 3 0 0 | 2 | 1152 | 8 | 8 | none |

*Table 10.* Qwen3-Next-80B training configurations.

| Qwen3-Next-80B | Pipeline Placement | BS | GBS | DP | EP | Recompute |
|---|---|---|---|---|---|---|
| `Naive` (OOM) | 6 6 6 6 6 6 6 6 | 2 | 1152 | 8 | 8 | layernorm |
| `Naive` | 5 6 6 6 6 6 7 6 | 2 | 1152 | 8 | 8 | layernorm |
| `LR` (OOM) | 5 7 7 7 7 7 7 1 | 2 | 1152 | 8 | 8 | layernorm |
| `LR` | 5 6 7 7 7 7 7 2 | 2 | 1152 | 8 | 8 | layernorm |
| AdaHC | 3 3 3 3 3 3 3 3 3
3 3 3 3 3 3 3 0 0 | 2 | 1152 | 8 | 8 | layernorm |

*Table 11.* Qwen3-235B training configurations.

| Qwen3-235B | Pipeline Placement | BS | GBS | DP | EP | Recompute |
|---|---|---|---|---|---|---|
| `Naive` | 9 9 9 10 10 10 10 9 9 9 | 1 | 640 | 8 | 8 | layernorm&Attn&MoE |
| `LR` | 9 9 9 10 10 10 10 10 10 7 | 1 | 640 | 8 | 8 | layernorm&Attn&MoE |
| AdaHC | 1 2 2 2 2 2 2 2 2 2
2 2 2 2 2 2 2 2 2 2
2 2 2 2 2 2 2 2 2 2
2 2 2 2 2 2 2 2 2 2
2 2 2 2 2 2 2 2 1 0 0 | 1 | 640 | 8 | 8 | layernorm&Attn&MoE |

