# OpenReview forum: "AdaHC: Accelerating Multi-Token Prediction with Adaptive Head Chunking with Pipeline Parallelism"
_ICML.cc/2026/Conference — ICML 2026 regular_

### Official Review · Reviewer_DavC · 2026-02-13

**Soundness:** 3
**Presentation:** 3
**Significance:** 3
**Originality:** 3
**Overall Recommendation:** 4
**Confidence:** 5

**Summary:**

AdaHC uses the concept of load balancing, splitting the MTP output header to achieve better pipeline parallelism scheduling. AdaHC splits the output heads into chunks and reassembles the chunks to generate balanced pipeline stages, and performs adaptive activation forwarding to preserve the numerical equivalence. 1.35× training throughput is achieved by AdaHC.

The main idea of AdaHC is Head Chunking, split and reassembles the output heads into a set of normalized chunks. This idea has been deployed on [1] [2] [3] for both single-GPU training and pipeline parallelism. Using such an idea for Multi-Token Prediction is something new and a reasonable extension with proper speedup.

For those reasons, we tend to weak accept.

[1] Luo C, Zhao J, Chen Z, et al. MINI-SEQUENCE TRANSFORMER: optimizing intermediate memory for long sequences training[C]//Proceedings of the 38th International Conference on Neural Information Processing Systems. 2024: 97299-97327.

[2] Luo Q, Li M, Zhao L, et al. StreamBP: Memory-Efficient Exact Backpropagation for Long Sequence Training of LLMs[J]. arXiv preprint arXiv:2506.03077, 2025.

[3] Yao J, Jacobs S A, Tanaka M, et al. Training ultra long context language model with fully pipelined distributed transformer[J]. Proceedings of Machine Learning and Systems, 2025, 7.

**Compliance With Llm Reviewing Policy:**

Affirmed.

**Final Justification:**

The authors have solved my concern; we are open to accepting it.

**Key Questions For Authors:**

How would adaHC handel different sequence lengths where the optimal chunk size might change across micro-batches?

**Strengths And Weaknesses:**

Strengths:

Soundness: AdaHC has three modules: CDT constructor, CDT Planner, and CDT Executor, which are clearly well-structured for pipeline parallelism. We are conviced that such a design is clear to demonstrate how it accelerates Multi-Token Prediction with Adaptive Head Chunking with Pipeline Parallelism. Load balance is also good and practical.


Weaknesses:

Presentation: The cost model for chunking size is unclear. Equation (3) simply says "solve for s such that the output head time matches the transformer layer time," However, the kernel execution time can vary with memory movement, gpu untilization, kernel shape. We recommend roofline analysis for detailed analysis between memory-intensive kernel and compute-intensive kernel.

Experiment. imited scope of pipeline schedules evaluated. The paper evaluates only 1F1B and interleaved 1F1B. More recent schedules like Zero Bubble (Qi et al., 2023) and DualPipe (Liu et al., 2024) are discussed but not experimentally compared. Since these schedules also target bubble reduction, the reader is left wondering whether AdaHC is complementary to or redundant with them.

---

> ### Author Rebuttal · Authors · 2026-03-30
>
> Thank you for your review and positive assessment. We are glad that you appreciate the hierarchical design of our method. We respond to your main concerns below.
>
> ## Q1. Novelty of Head Chunking
> We agree that chunking itself is not new. Prior work splits sequences into chunks to improve long-sequence training efficiency. However, AdaHC is novel in both the problem setting and the system role.
>
> - **Different problem setting**. AdaHC targets a previously underexplored bottleneck: MTP-induced tail-stage imbalance in PP, caused by multiple output heads concentrating at the end of the model. This differs from the long-sequence memory/execution problems in [1][2][3].
> - **Different role of chunking**. In AdaHC, chunking is not mainly an execution/memory technique. It converts OH computation into scheduling units matching TF blocks, so that the PP stage partition can be rebalanced.
> - **New scheduling/execution framework**. Beyond chunking, AdaHC introduces the CDT abstraction, a dependency-aware planner, and a correctness-preserving executor with activation piggybacking and loss aggregation. These make chunked OH computation usable for MTP+PP training.
> So we agree the novelty is not “inventing chunking,” but turning chunking into a unified scheduling/execution framework for the MTP-specific PP imbalance problem. We will clarify this contribution boundary in the final version.
>
> ## Q2. Details of Cost Model
> We agree that Eq. (3) is a lightweight target, not a strict analytical performance model. Our goal is not exact runtime prediction, but obtaining a chunk size accurate enough for OH-splited PP scheduling.
> - For TF block, we do not rely on a hand-crafted analytical latency model, since its runtime is determined by multiple heterogeneous kernels and is difficult to predict accurately. Instead, we use an existing transformer performance analysis tool (llm-analysis) to estimate TF-block cost.
> - For OH, the dominant cost is the projection GEMM(`hidden_state @ output_head.weight`), which accounts for more than 95% of the OH time, making a simple sequence-length-based model sufficiently accurate.
> - Before training, we calibrate with profiling (e.g., Nsight Systems), and use lightweight runtime timers to verify the chunk size.
>
> Appendix B.2 shows that the chosen chunk size is already accurate enough in practice, and throughput is not highly sensitive to small chunk-size deviations. We agree that a roofline-style analysis could provide more kernel-level characterization. However, AdaHC only needs coarse-grained latency matching between OH chunks and TF blocks, for which empirical profiling is sufficient. We will emphasize this in the final version.
> ## Q3. Comparison with More PP Schedules
> We agree that these suggested pipeline scheduling strategies are valuable. However, AdaHC mainly targets **tail-heavy stages caused by output-head structure**, which lead to **imbalance bubbles and schedule bubbles** in PP. This bottleneck is not the same as the optimization target of many existing PP scheduling strategies, so they are not direct replacement baselines for the bottleneck studied here.
>
> Specifically, Zero Bubble and DualPipe generally assume stage computation is relatively balanced. Their main focus is reducing warmup/cooldown bubbles or improving forward/backward overlap. When multiple output heads break the stage-balance assumption, such schedule-level optimizations usually cannot directly eliminate architecture-induced stage imbalance.
>
> In contrast, AdaHC reconstructs the output heads into schedulable chunks, mitigating **architecture-induced bubbles** and enabling finer interleaved schedules. AdaHC is thus **orthogonal and complementary** to them.
>
> In the final version, we will further discuss the distinction between AdaHC and existing PP scheduling strategies and emphasize this complementarity/composability, to avoid giving readers the impression that these methods are direct substitutes.
> ## Q4. Different Sequence Lengths across Micro-Batches
> Great question. Imbalanced micro-batches caused by variable sequence lengths are common in PP for long-sequence training. In Section 2.2, we define this as **data bubble**, and current long-context training already applies bucketing / packing / length balancing so that micro-batches within a run are relatively similar in cost.
>
> AdaHC is designed for this common setting and focuses on **schedule bubbles and imbalance bubbles** related to model architecture. Therefore, it uses a chunk size calibrated for the target training configuration rather than re-optimizing chunk size for every individual micro-batch.
>
> As noted in Q2, throughput is robust to small chunk size deviations. And existing data-bubble solutions are **orthogonal**: they first mitigate length-induced bubbles, then AdaHC further addresses structural OH imbalance. Extending AdaHC to fully dynamic per-micro-batch chunk sizing is an interesting future direction.

---

> > ### Author Rebuttal · Reviewer_DavC · 2026-04-03
> >
> > We are glad that the author will clarify this contribution boundary in the final version. We are suggesting accepting this paper after revision.
> >
> > Although we would keep our overall recommendation because it is rated for the original paper, not the revision, we are still open to accepting it.

---

> > > ### Author Response · Authors · 2026-04-04
> > >
> > > Thank you very much for your follow-up and for marking our response as **“Fully resolved.”** We sincerely appreciate your positive feedback and your note that you are suggesting acceptance after revision.
> > >
> > > We also understand and respect your point that the original overall recommendation was made for the submitted version rather than a revised one.
> > >
> > > If appropriate, we would be grateful if you could consider whether a small score update would better reflect your post-rebuttal assessment for the AC/PC. Since you indicated that the concerns are fully resolved, we worry that leaving the score unchanged may understate the effect of the rebuttal.
> > >
> > > In any case, thank you again for the thoughtful discussion. We will incorporate the promised clarifications in the final version, including the contribution boundary, the role of the cost model, and the positioning of AdaHC relative to other PP scheduling methods.

---

### Official Review · Reviewer_Hx2L · 2026-02-25

**Soundness:** 3
**Presentation:** 3
**Significance:** 3
**Originality:** 3
**Overall Recommendation:** 4
**Confidence:** 4

**Summary:**

This work proposes AdaHC, an adaptive pipeline parallel scheduling framework for training large language models (LLMs) equipped with Multi-Token Prediction (MTP) modules.

Core ideas:

- It exploits the parallel nature of MTP output heads (there are no dependencies across different heads, and each head’s computation can be partitioned along the sequence dimension);

- It decomposes output heads into “head chunks” whose granularity is comparable to transformer layers, and treats these chunks as unified scheduling units;

- It introduces a Chunk Descriptor Table (CDT) and a hierarchical scheduler to map scheduling units onto pipeline stages, while also applying operator fusion;

- During execution, it employs adaptive activation forwarding and loss aggregation to minimize communication overhead.

**Compliance With Llm Reviewing Policy:**

Affirmed.

**Final Justification:**

As mentioned in final rebuttal, although the author has made some explanations, there are still parts that have not been resolved, thus I believe my current score is appropriate.

**Key Questions For Authors:**

Please refer to weaknesses.

**Limitations:**

Please refer to weaknesses.

**Strengths And Weaknesses:**

## **Strengths**
The work targets an important but underexplored challenge in pipeline parallel training with MTP modules. While existing PP methods (e.g., Megatron-LM, GPipe, 1F1B/Interleave) focus on layer-level scheduling, they rarely address stage imbalance caused by MTP heads. This makes the contribution complementary to prior PP research by explicitly tackling the imbalance bubble introduced by MTP outputs.

It proposes practical communication optimizations. Adaptive activation piggybacking reduces unnecessary data transfers by forwarding activations only when needed, and bit-wise loss aggregation minimizes high-frequency small-message communication while preserving numerical equivalence. These mechanisms are well suited for bandwidth- and latency-sensitive PP systems.

Evaluation spans multiple model scales and MTP configurations, demonstrating applicability across different settings rather than a single narrow case.



## **Weaknesses**
1. **Limited evaluation of model quality and convergence.**
   The experiments focus primarily on throughput and stage load balance, with little analysis of:

   - training convergence behavior,

   - final perplexity or downstream performance,

   - training stability.

   As a result, the impact of the proposed scheduling on model quality remains insufficiently characterized.


2. **Insufficient comparison with state-of-the-art PP scheduling methods.** Although the paper compares against Naive, Vocabulary Parallel, and Layer Redistribution, it does not benchmark against more advanced pipeline parallelism techniques such as Dynamic Pipeline Scheduling (e.g., token-dimension scheduling methods like TeraPipe) or Zero-Bubble Scheduling / forward-backward overlap strategies. These approaches propose more fine-grained solutions for pipeline bubble reduction and dynamic scheduling, yet are absent from the baseline comparisons.

3. **Insufficient analysis of communication bottlenecks at scale.**
While activation piggybacking and loss packing reduce communication overhead, the paper lacks detailed system-level profiling under cross-node (RoCE/RDMA) high-latency environments. Missing analyses include:

   - latency and bandwidth sensitivity of the proposed mechanisms,

   - behavior of hidden-state forwarding under dense vs. sparse network topologies,

   - GPU timeline utilization analysis.

---

> ### Author Rebuttal · Authors · 2026-03-30
>
> Thank you for the careful review and positive feedback. Below we respond point-by-point to the weaknesses you raised.
> ## Q1. Model Quality and Convergence
> We fully agree that training convergence and stability are critical criteria for evaluating the practical usefulness of AdaHC. AdaHC is designed to optimize the pipeline bubbles introduced by output heads **without changing the original training objective and numerical semantics**, based on the following guarantees:
> - For already computed hidden states, the output head and its corresponding cross-entropy loss can be computed independently along the sequence dimension. AdaHC only performs **lossless partitioning and rescheduling** of the output-head/loss computation, and does not change the Transformer backbone’s modeling process.
> - Loss aggregation uses the **bit-wise transparent** packing/unpacking described in the paper, which does not change the actual numerical representation of the loss.
> - The **gradients** contributed by all head chunks to the shared output-head parameters are accurately aggregated before parameter updates, and therefore remain consistent with the original training procedure.
>
> In addition, we have provided a convergence verification experiment in **Appendix B.1 (Accuracy Verification)**. Under a controlled setting, the training loss curves of AdaHC and the Naive baseline are nearly identical, and we do not observe systematic loss shift or training instability.
>
> AdaHC has already been deployed in our production clusters and has served more than **100** jobs. We agree with your suggestion that the presentation of final model quality is still not sufficient in the current version. In the final version, we will provide a more complete discussion of model quality, including a clearer presentation of convergence consistency results and, as much as possible, perplexity / downstream-task performance.
> ## Q2. Comparison with More Pipeline Scheduling Strategies
> We agree that these suggested pipeline scheduling strategies are valuable. However, we would like to clarify that AdaHC mainly targets the **tail-heavy stages caused by output-head structure**, which in turn lead to **imbalance bubbles and schedule bubbles** in pipeline parallelism. This bottleneck is not exactly the same as the optimization target of many existing PP scheduling strategies, so these strategies are not fair replacement baselines for the bottleneck studied in this work.
>
> Concretely, the Dynamic Pipeline Scheduling strategies you mentioned (e.g., TeraPipe) and optimizations such as forward-backward overlap typically assume that stage computation is already relatively balanced. They mainly reduce warmup/cooldown bubbles or improve forward/backward overlap efficiency. However, when multiple output heads break the stage-balance assumption, such schedule-level optimizations usually cannot directly eliminate the architecture-induced stage imbalance.
>
> In contrast, AdaHC reconstructs the output heads concentrated in the tail stage into schedulable chunks, thereby fundamentally mitigating the architecture-induced bubble and also enabling finer-grained interleaved schedules. In this sense, AdaHC is **orthogonal and complementary** with them.
> In the final version, we will further discuss the distinction between AdaHC and existing PP scheduling strategies, and more clearly emphasize this complementarity/composability, to avoid giving readers the impression that these methods are direct strategies for each other.
> ## Q3. Analysis of Communication Bottlenecks at Scale
> We agree that the current main paper does not provide sufficiently detailed system-level network sensitivity analysis. This omission is primarily due to the following reasons:
> - AdaHC **fully reuses** the communication path already used in PP, and the communication volume introduced by AdaHC is of the same order as standard PP communication. Therefore, AdaHC adds only extra payload on the existing P2P path rather than introducing a new synchronization bottleneck, and this path itself has already been systematically validated in the state-of-the-art PP systems (e.g., Megatron-LM).
> - We have already provided a quantitative analysis in **Appendix B.4 (Piggybacking Pipeline Communication Overhead)**. In the settings used in this paper, the maximum additional piggyback payload is about **33.6MB**, corresponding to a theoretical transmission time of only **671us**.
> - This communication reuses the original pipeline P2P communication path, and under 1F1B / interleaved 1F1B scheduling it can usually **overlap with the computation** of neighboring micro-batches. As a result, its actual end-to-end visible overhead is typically much lower than the theoretical upper bound.
> - All experiments in this paper are also conducted in real multi-node environments with **cross-node RoCE communication**, rather than being limited to single-node settings.

---

> > ### Author Rebuttal · Reviewer_Hx2L · 2026-04-03
> >
> > The rebuttal provides useful clarifications but does not sufficiently address the core concerns regarding empirical validation and fairness of comparisons. In particular, absence of comparisons with advanced PP scheduling methods, and limited system-level analysis at scale leave the main claims insufficiently supported. Therefore, I maintain my original score.

---

> > > ### Author Response · Authors · 2026-04-04
> > >
> > > Thank you again for the follow-up. We understand your concern that the current paper does not yet make the empirical support sufficiently explicit, especially regarding advanced PP schedulers and system-level evidence.
> > >
> > > ## AdaHC and Advanced PP Scheduling Methods are Orthogonal.
> > > On the comparison with advanced PP scheduling methods, we would like to emphasize one concrete point beyond our previous rebuttal: **AdaHC can be applied on top of those schedulers, including Zero-Bubble-style schedules**. To make this more explicit, we provide an anonymous supplementary schedule figure here: [AdaHC with 1F1B & ZeroBubble.pdf](https://anonymous.4open.science/r/ICML2026-5821/AdaHC%20with%201F1B%20&%20ZeroBubble.pdf).
> > >
> > > The figure shows two effects clearly. First, after AdaHC splits and redistributes the output-head computation, it removes the **MTP-induced tail imbalance bubble**, which is the core bottleneck studied in this work. Second, once the output heads are converted into finer-grained chunks, the scheduler has more flexible units to arrange, which makes interleaved / Zero-Bubble-style scheduling **more effective at reducing the remaining schedule bubble**. In other words, AdaHC does not conflict with advanced PP schedulers; it improves the model partitioning granularity and stage balance so that those schedulers can work under a better-shaped pipeline.
> > >
> > > We agree that this composability was not presented strongly enough in the submission. In the final version, we will add this AdaHC+Zero-Bubble scheduling figure and an explicit discussion to better position AdaHC as **complementary to**, rather than competing with, prior PP scheduling techniques.
> > >
> > > ## Limited System-level Analysis at Scale
> > > On the system-analysis side, we also agree that the current main paper could surface the evidence more clearly. At the same time, we want to clarify that the experiments are already conducted in **real multi-node environments with cross-node RoCE communication**, not just single-node settings, and the tested configurations already cover the production-relevant scaling range for the corresponding jobs. Our scaling analysis is not limited to one axis: within DP groups, we evaluate across **model size, PP size, and EP size**, and these settings already reach the normal upper range used in our production training tasks. Further increasing PP size beyond that operating region would likely provide only limited additional insight into the specific bottleneck studied here.
> > >
> > > In addition, several relevant system analyses are already included in the paper/appendix, although admittedly not highlighted enough:
> > > - Section 5.3 reports the **stage-load balance CDF**, showing AdaHC achieves the most uniform workload distribution;
> > > - Appendix B.3 reports **memory behavior** across pipeline stages;
> > > - Appendix B.4 quantifies the **piggyback communication overhead** and explains that it reuses the existing PP P2P path rather than introducing a new synchronization path;
> > > - Appendix C provides a **quantitative communication/synchronization analysis** for VP.
> > >
> > > We appreciate this suggestion and will make these system results more visible in the final version, together with the new [AdaHC+Zero-Bubble figure](https://anonymous.4open.science/r/ICML2026-5821/AdaHC%20with%201F1B%20&%20ZeroBubble.pdf), so that the paper more clearly demonstrates both the empirical effectiveness of AdaHC and its compatibility with existing advanced PP schedulers.

---

### Official Review · Reviewer_G2iS · 2026-03-22

**Soundness:** 4
**Presentation:** 3
**Significance:** 3
**Originality:** 3
**Overall Recommendation:** 5
**Confidence:** 1

**Summary:**

This paper studies the pipeline parallelism efficiency problem in LLMs with MTP architectures. When training with pipeline parallelism, MTP output heads concentrate on the tail pipeline stage, creating severe load imbalance and pipeline bubbles. The paper identifies two key properties of MTP output heads: independence across heads and partitionability along the sequence dimension — and proposes AdaHC, a framework that exploits these properties to distribute output head computation across pipeline stages via a unified scheduling abstraction (CDT), reducing both imbalance and schedule bubbles.

**Compliance With Llm Reviewing Policy:**

Affirmed.

**Key Questions For Authors:**

Questions:
* Could you explain how the backward pass of the OH is handled, specifically concerning the sequence-dimension splitting and the replication of OH weights across multiple stages? I assume the backward pass is handled correctly since the training loss curve is successfully reproduced as shown in Appendix Figure 7, but I am curious about the underlying mechanism.
* Could you discuss the rationale for prioritizing OH when breaking ties at the same dependency level? Would alternative ordering strategies, such as prioritizing TF blocks first or random ordering, yield different or better results?
* How would the effectiveness of AdaHC change with longer context lengths?

Minor issue:
* "The hidden states are the passed to the first MTP block." -> "are then passed to"

**Limitations:**

The paper does not include a dedicated limitations section. The scaling experiment (Section 5.4) uses a truncated backbone to control the hardware configuration across different MTP depths, rather than evaluating on a full-scale production model with D>1. Overall, the limitations discussion could be more explicit but the omission is not a significant concern.

**Strengths And Weaknesses:**

Strengths:
* The paper is well-written and the figures are well-designed.
* The problem is clearly motivated and the high-level solution is intuitive.
* The experiments are comprehensive, including accuracy verification and ablations on peak GPU memory, throughput, and chunk size sensitivity in the appendix.
* The results are convincing, with improvements over baselines that grow as MTP depth increases.

Weakness:
The paper is technically solid and I have no major concerns regarding correctness or experimental design. The remaining open questions I have are primarily about the CDT Planner's heuristic choices and the backward pass through distributed OH chunks, which I have detailed in the questions section below.

---

> ### Author Rebuttal · Authors · 2026-03-30
>
> Thank you for the positive assessment and for the thoughtful questions. We are glad that you found the paper technically solid. Below we clarify the three points in more implementation detail. These mechanisms are already used in our system and we will also make them explicit in the final version.
>
> ## Q1. Details of OH Correctness Guarantee
>
> - **From the perspective of computation:** the cross-entropy loss of each token depends only on that token's logits and does not depend on other sequence positions. Therefore, after splitting the input into chunks along the sequence dimension, the total loss of the full OH can be represented exactly as the sum of token-wise losses over all chunks, without introducing any approximation error. This also means that, in backward, each OH chunk produces gradients only for its own token span, and these chunk-wise gradients together exactly recover the gradient of the original unsplit OH.
>
> - **From the perspective of parameter consistency during training:** AdaHC places replicas of the same shared OH parameters on multiple stages. At the same time, for the stages with OH in each PP group, we create a dedicated communication group. Before the parameter update, AdaHC uses all-reduce to synchronize the total gradients of these OH replicas, ensuring that all replicas are updated consistently and correctly.
>
> - **Initialization / checkpoint:** in implementation, if we load a checkpoint that was not produced with AdaHC, we directly copy the OH parameters from the last stage to the OH replicas on non-last stages. If training starts from scratch, we synchronize the OH parameters across stages before training begins.
>
> ---
>
> ## Q2. Block Scheduling Priority
>
> To make the OH priority choice more explicit, we summarize the trade-offs of different tie-breaking strategies below. The key consideration is **communication locality**: placing OH closer to its dependent TF reduces the need for cross-stage hidden-state forwarding, while alternative strategies may preserve correctness but typically incur more communication overhead or less stable mappings.
>
> | Tie-breaking strategy | Advantages | Disadvantages |
> |---|---|---|
> | **OH-first**: Places OH as close as possible to its dependent TF | ● Minimizes cross-stage hidden-state forwarding ● Communication-aware and well aligned with data locality ● Stable and simple default heuristic | ● May not be strictly optimal under some extreme slot layouts |
> | **TF-first**: Places TF as close as possible to its dependent TF | ● Does not affect correctness ● Simple scheduling rule | ● Tends to push OH into later slots ● Increase the likelihood of OH forwarding and thus extra communication overhead |
> | **Random**: Random placement | ● Does not affect correctness ● Serves as a simple unbiased baseline | ● Lacks data-locality awareness ● May place OH farther downstream ● Lead to unstable mappings and higher forwarding overhead |
>
> ---
>
> ## Q3. Effectiveness under Longer Context Length
>
> AdaHC is compatible with long-context training and is **orthogonal** to long-context setting such as context parallel or sequence parallel and so on. Its benefit is mainly determined by the relative compute share of OH versus TF, rather than by context length alone.
>
> - **AdaHC directly targets OH-induced tail imbalance.** Under PP, concentrating OH on the tail stages causes stage imbalance and thus reduces pipeline efficiency. Therefore, the gain of AdaHC depends directly on whether OH remains sufficiently "heavy" relative to TF.
>
> - **Under longer sequences**, both TF and OH costs increase, and TF with full attention may grow faster. In particular, TF can be approximated as `FullAttention(S²) + FFN(S)`, while OH is closer to `OH(S)`. Therefore, simply increasing sequence length may reduce the relative contribution of OH compared with TF, which can weaken the benefit of AdaHC. In practical long-context training, context parallelism and sequence parallelism are often also used, so the actual TF bottleneck is influenced by the overall parallel configuration.
>
> > **Note:** As long as OH-induced tail imbalance is not negligible, AdaHC can provide benefits.
>
> ---
>
> ## Q4. Typo and Limitations
>
> We thank the reviewer for pointing out these minor typos. We will correct them in the final version and conduct a thorough proofreading of the entire manuscript to eliminate any remaining errors.
>
> Finally, we agree that the limitations can be made more explicit, and we will add a brief limitations discussion in the final version, including the truncated-backbone scaling setup and expected behavior for larger/full-scale multi-MTP deployments.

---

### Decision · Program_Chairs · 2026-04-30

**Decision:**

Accept (regular)

**Comment:**

The paper introduces AdaHC (Adaptive Head Chunking), a framework designed to optimize pipeline parallelism (PP) for Large Language Models using Multi-Token Prediction (MTP) architectures. The core problem addressed is the "tail-heavy" imbalance caused by MTP blocks, which typically concentrate output head (OH) computation at the final pipeline stage, leading to significant idle bubbles. AdaHC leverages the independence of MTP heads and the partitionability of the sequence dimension to split output heads into "head chunks." These chunks are then redistributed across pipeline stages using a Chunk Descriptor Table (CDT) to balance the load. The system includes adaptive activation forwarding to maintain numerical equivalence ad minimize communication overhead.

AdaHC offers a timely and effective solution to a problem in distributed LLM training. The authors have demonstrated significant performance gains without sacrificing numerical accuracy. Although some reviewers felt the empirical comparison with advanced schedulers could be more exhaustive in the main text, the rebuttal successfully demonstrated the framework's compatibility and orthogonality. Given the solid technical contribution and the positive consensus among reviewers, the paper is recommended for acceptance.